# Vaccination against SARS-CoV-2 in Haemodialysis Patients: Spike’s Ab Response and the Influence of BMI and Age

**DOI:** 10.3390/ijerph191610091

**Published:** 2022-08-15

**Authors:** Pedro Ponce, Ricardo Peralta, Carla Felix, Carla Pinto, Bruno Pinto, João Fazendeiro Matos

**Affiliations:** 1Country Medical, NephroCare Portugal, Fresenius Medical Care Portugal, 1750-233 Lisboa, Portugal; 2Direção de Enfermagem, NephroCare Portugal, Fresenius Medical Care Portugal, 4470-573 Porto, Portugal

**Keywords:** SARS-CoV-2, COVID-19, vaccine, antibody, immunity, dialysis

## Abstract

Patients with chronic kidney disease (CKD-5D) in dialysis have been associated with higher rates of SARS-CoV-2 infection. Objective: To identify the CKD-5D patients’ immune system behavior regarding the Pfizer-BioNTech (BNT162b2 mRNA) vaccine (Comirnaty©). This was a multicenter study carried out in 38 dialysis units in NephroCare Portugal. Eligible patients from two cohorts—one composed of completely vaccinated patients with Comirnaty© (vaccinated group) against a second cohort of patients who recovered from SARS-CoV-2 infection (control group)—were selected through representative sampling for each cohort. Humoral response was assessed at 3 (t0) and 6 months (t1) after complete vaccination and, in the control group, 6 months after COVID-19 recovery. In the vaccinated group, at t0, the median anti-Spike IgG level was 1120 AU/mL and, at t1, all participants’ antibody level decreased to a median of 455 AU/mL. In the control group, the median serum SARS-CoV-2 antibodies level was 1836 AU/mL. In the vaccinated group, at t0, patients < 70 years presented a significantly (*p* = 0.002) higher level of anti-Spike IgG titres. In contrast, older patients from the control group presented a significantly (*p* = 0.038) higher IgG. No correlation was found between age and anti-Spike IgG antibodies level in any of the studied groups. Patients with a higher body mass index showed a greater immune response in both the vaccinated and control group, although without significance. We concluded that, in the vaccinated group, elderly patients developed a lower immune response than younger patients and the levels of anti-Spike IgG antibodies declined faster between t0 and t1, while in the control group, the oldest and overweight patients developed the best humoral response.

## 1. Introduction

The novel coronavirus disease 2019 (COVID-19) caused by severe acute respiratory syndrome coronavirus 2 (SARS-CoV-2) has been associated with causing high incidence, mortality, and fatality rates in people with chronic kidney disease in stage 5 (CKD-5D) [1,2,3,4] in a regular dialysis program. According to the literature, this group of patients is five to sixteen times more likely to be infected with SARS-CoV-2 and has nearly a 30 to 130% higher risk of death than the general population [5]. Thus, prioritizing this particular group of patients for vaccination was a common strategy in several countries, like Portugal [6].

Current vaccination programs include the Moderna (mRNA-1273), Pfizer-BioNTech (BNT162b2 mRNA) Comirnaty©, Oxford/AstraZeneca (ChAdOx1 nCoV-19), and Johnson & Johnson COVID-19 (Ad26.COV2.S) vaccines, which have all been shown to induce both humoral and cellular immune responses [7,8,9] against the Spike protein of the SARS-CoV-2 virus, protecting individuals from the risk of subsequent infection. The vaccination regime for the Comirnaty© vaccine (composed of two doses) showed a 95% induced protection against COVID-19 in the general population. However, CKD-5D patients were not included in the clinical trials, thus there are no data on the efficacy for this group of patients [10,11].

With the establishment and progression of chronic kidney disease, it has been confirmed that patients with this infirmity have a pro-inflammatory condition that leads to an immune system disorder. So, this chronic condition makes patients more susceptible to infections, as well as to virus-associated cancer, and causes them to respond poorly to the standard vaccination protocols, as previously observed with hepatitis B [12] and influenza vaccinations [13], which revealed lower rates of antibody seroconversion and reduced durability. Moreover, in dialysis patients, CD8+ T cells are diminished in both number and function. Because both arms of the immune system are usually impaired, this could explain the increased susceptibility of these patients to viral and bacterial infections and their poor vaccination responses [14].

It is of critical importance to identify the behavior of the CKD-5D patients’ immune system regarding the Comirnaty© vaccine in order to anticipate future risks and to provide sufficient information to plan the future strategies to combat COVID-19 in this population.

The objective of this study is to evaluate the immune response of CKD-5D patients to the Comirnaty^®^ vaccine after a period of 3 and 6 months after completing the vaccination regime, and compare it with that of non-vaccinated patients infected with SARS-CoV-2.

## 2. Materials and Methods

### 2.1. Participants’ Inclusion

Enrolled patients were selected from 38 NephroCare Portugal (Fresenius Medical Care Portugal) dialysis units into two cohorts: one composed of completely vaccinated patients with Comirnaty^®^ (vaccinated group) against a second cohort of patients who recovered from COVID-19 (control group), confirmed by the RT-PCR laboratory technique. The inclusion criteria were patients who completed vaccination in our dialysis clinic network until 20 February 2021 (vaccination is considered completed 7 days after the second immunization), and patients who had an infection in the sixth (±1) month prior to blood sample collection. Regarding vaccinated patients, only data from patients with valid outcomes in both blood sample periods was considered. Out of the 3672 eligible patients, we created three age subgroups according to the following criteria: <60 years; 60 to 70 years; and >70 years old. Three subgroups were also created for the body mass index (BMI) variable according to the quartiles.

### 2.2. Sample Collection

Through a randomization process, a representative sample for each group was obtained with a 95% confidence interval and 5% of sample error, a process conducted according to Bartlett JE, et al. [15]. In order to achieve representativeness for the vaccinated population, a sample of 385 patients was considered, including 10% to accommodate dropouts during the 6 months of follow-up. The same procedures were followed to calculate the representative sample (205) of the control group.

### 2.3. Anti-Spike IgG Level Quantification

For this purpose, with regard to the vaccinated group, we performed blood sampling in order to quantify the specific Spike antibody IgG (titre of s-Ab) of the vaccine-induced immune responses in two different moments, after 90 and 180 days (±15 days) of completing the vaccination, and with regard to the control group, we performed blood sampling in order to quantify the specific Spike antibody IgG (titre of s-Ab) of the SARS-CoV-2 response to COVID-19 after 180 days (±15 days) of a positive test for SARS-CoV-2. As previously suggested, the threshold for seroconversion was 50 arbitrary units per millilitre (AU/mL) [16].

A total of mL of blood was collected in the dialysis unit immediately at the beginning of treatment when scheduled. Immunogenicity was assessed by chemiluminescence immunoassay (CLIA) to quantify IgG antibodies Anti-SARS-CoV-2 IgG Spike.

### 2.4. Measure of Body Mass Index

Body mass index was obtained by bioimpedance spectroscopy measurements of the body composition with the BCM(R) (Body Composition Monitor—Fresenius Medical Care) and was defined as the dry weight (kg) divided by height in square meters.

### 2.5. Statistical Analysis

Categorial variables were reported as frequencies and percentages. Continuous variables were reported as median and interquartile range (IQR). For paired samples and when the assumption of normality was not verified, the Wilcoxon test was used. For the comparison of different groups (vaccinated versus infected patients), the Mann–Whitney U test was used, when appropriate. The results were considered significant when they had a *p* < 0.05. All of the statistical analyses were performed using SPSS (version 23; IBM, Armonk, NY, USA).

## 3. Results

This observational, prospective, and multicentre study had a cohort of 4609 patients on haemodialysis (HD) distributed in 38 clinics. A total of 3672 were scheduled for vaccination against SARS-CoV-2 between January and February 2021. From that group, 385 participants with no evidence of previous exposure to SARS-CoV-2 were randomized to the study (vaccinated group) (Figure 1). A total of 695 tested positive for SARS-CoV-2 in RT-PCR laboratory technique tests from December 2020 to February 2021, 205 of whom were enrolled for the “control group”.

Eighty-six participants were lost during the follow-up for the reasons described in Figure 1. In the vaccinated group, five participants were excluded owing to later infection by SARS-CoV-2; even so, only one had dyspnoea, diabetic decompensation, and was hospitalized for 6 days. The remaining four patients did not present symptoms and were only identified after performing routine tests. Therefore, none of those patients developed severe symptoms requiring intensive care admission or died.

The median age was 68 years old (range (19–95), IQR = 57–79) with mostly males (*n* = 297, 58.90%), and the control group was older (*p* = 0.042) (Table 1).

### 3.1. Humoral Response of Vaccinated Patients after 3 and 6 Months

The mean interval between the end of completion of the vaccination program with the Comirnaty© vaccine and the first blood sampling was 3.63 ± 0.06 (range, 3.03–3.87) months (t0). Overall, at t0, the median anti-Spike IgG level was 1120 (IQR = 493–2805) AU/mL. Eight patients (2.49%) were weak responders with a median of 32 (range 11–50) AU/mL. No non-responders were observed. Among those eight, four (50%) were diabetics, older, and heavier (median 76, IQR = 62–85 years and BMI 27.95, IQR = 25.8–32.70 Kg/m^2^, respectively) than the respective overall group; however, they had been on haemodialysis for a shorter period of time. The second blood sampling was performed at a mean of 6.64 ± 0.10 (range 6.53–7.10) months (t1). All participants decreased their antibody level to a median of 455 (IQR = 189–967) AU/mL and the weak responders increased to 21 (6.54%). The differentiating factor present in these participants was again the dialysis vintage, but in this case, these patients were on dialysis for a longer time when compared with the others.

### 3.2. Humoral Response of COVID-19-Recovered Patients (Control Group)

In the 183 patients recovered from COVID-19 (control group), the mean time from COVID-19 diagnosis to baseline blood sampling was 5.18 ± 0.94 (range = 4.4–9.23) months. Median serum SARS-CoV-2 antibodies level was 1836 (IQR = 749–5168) AU/mL. Out of those patients, four (2.18%) were weak responders with a median of 34 (range 21–50) AU/mL. Notably, these patients are younger, but have been on HD for much longer (median 57, IQR = 48–74 years and vintage dialysis 262, IQR = 149–314 months, respectively) than the others.

### 3.3. Anti-Spike Antibody Response between Groups

After 6 months of complete vaccination (t1), median anti-Spike IgG antibodies were significantly lower, at 455 (IQR = 189–967) AU/mL, when compared with t0 (median 1120 (IQR = 493–2805) AU/mL) (Wilcoxom = −15.597; *p* < 0.001). However, comparing the vaccinated group at t0 with the control group (recovered from COVID-19), the humoral response was significantly higher in the control group (Mann–Whitney = 13391; *p* < 0.001), with a median concentration of 1836 (IQR = 749–5168) AU/mL even after 5.18 months of seroconversion (Table 2).

In the vaccinated group at t0, humoral response in older patients (>70 years) was lower, at 874 (IQR = 351–2053) AU/mL, when compared with younger patients (≤60 years). However, in the control group, the immune response was the opposite. In this group, patients with >70 years had a better response (2096 (IQR = 890–9849) AU/mL) than in the remaining age subgroups (Table 3).

To assess the response of elderly HD patients, they were partitioned into two age subgroups, at <70 and ≥70 years. In the vaccinated group at t0, the patients with <70 years presented a significantly (*p* = 0.002) higher level of anti-Spike IgG titres (median 1380, IQR = 641–3021 AU/mL) than the older ones (Table 4). In contrast, the patients with ≥70 years from the control group presented a significantly (*p* = 0.038) higher IgG when compared with the <70 years subgroup (median of 2096, IQR = 882–9872 AU/mL versus median of 1807, IQR = 598–3945). However, we found no correlation between age and anti-Spike IgG antibodies level in any of the studied groups.

We stratified the BMI into three subgroups according to quartiles, <23, 23–28, and >28 Kg/m^2^. Therefore, we decided to determine the median anti-Spike IgG in each patient group (vaccinated and control group). The subgroup with >28 kg/m^2^ showed a higher humoral response, particularly in the control group, with a median concentration of 2485 [IQR = 818–5117] AU/mL (Table 3).

The weight variable was also partitioned into two subgroups, <30 and ≥30 kg/m^2^. It was observed that, in both groups, in the vaccinated group (t0, t1) and in the control group, overweight patients had higher levels of anti-Spike IgG antibodies than in the <30 kg/m^2^ subgroup (Table 4), but without statistical significance (*p* = 0.975, *p* = 0.889, and *p* = 0.689, respectively). Similarly to the weight variable, we also did not find a correlation between BMI and the anti-Spike IgG antibodies levels. More details are available in Appendix A.

When we crossed these two factors, age with BMI, we found that older and heavier patients developed higher antibodies levels in the control group (Figure 2). More details are available in Appendix A.

## 4. Discussion

In the present study, we describe the kinetics of humoral response after Comirnaty^®^ vaccination in patients with CKD-5D after 3 and 6 months compared with the control group (patients recovered from COVID-19). To the best of our knowledge, this is the first study comparing humoral response in patients under a regular haemodialysis program, with the above characteristics.

Our results showed that, after 3 months (109 days) of the second Comirnaty© vaccine dose (complete vaccination), a large majority (97.51%) of patients with CKD-5D maintain positivity for anti-Spike IgG, which was not observed in non-responders. This finding is similar to previous published data in which, 36 days after the second immunization dose, 93.4% of the HD patients were seropositive with a median level of 1599 AU/mL [17].

However, the results reveal that this group had a 59.38% decline in antibody level in just 90 days of interval between t0 and t1, decreasing its protection degree. These results are in line with other studies in which participants also lost titre levels three months after vaccination [18] and, in another study enrolling 41 HD patients, it was concluded that the seroconversion rate decreases from 98% at 1 month to 65.80% at 6 months after the second dose [19]. A study conducted by Labriola et al. [20] also noted an identical decline (median of 65.6%) in all SARS-CoV-2-positive patients from 30 days after seroconversion through the following 3 months. This reduction may be associated with the uremic condition in patients undergoing hemodialysis and, therefore, plays an adverse role in the development of adaptive immune responses [14,21]. This factor also seems to contribute to the lower humoral response reported in recent studies [16,21,22], which showed that HD patients develop impaired humoral responses after vaccine dose when compared with the general population. Despite this significant reduction, we observed that 93.46% of patients after 180 days were still considered positive, which was not observed in any participants with seroconversion to negative.

The maintenance of antibodies from patients in the control group was significantly higher than that of those in the vaccinated group (t0 and t1), even after a median of 158 days. On the other hand, the response rate was also higher (97.82%) in the control group.

The findings on the partitioning of participants by age revealed that the immune response was lower in the oldest vaccinated group (≥70 years), when compared with age-matched participants in the control group. Previous studies have also concluded that age was an important factor in the humoral response following the Comirnaty© vaccine [16,22,23,24]. Age is another factor that may explain, at least in part, the lower humoral response in patients on dialysis. In opposition, our results showed that SARS-CoV-2 infection induces a significantly stronger immune response, particularly in older patients. We did not find an explanation for these findings, but in our interpretation, it may be attributable to the high severity of COVID-19 and, consequently, to the higher immune response in patients with multiple comorbidities. Thus, elderly patients presented a more severe COVID-19 clinical setup, with an increased risk of developing dyspnoea or pneumonia and mortality in HD patients that was 19.6% higher as compared with the non-elderly group [25]. Some studies have shown that HD patients have a robust and sustained antibody response when they have recovered from severe COVID-19, and these levels can be maintained over time [26,27].

It was observed that heavier patients that were considered obese (BMI ≥ 30 Kg/m^2^) showed a greater immune response in both the vaccinated and control group; however, we did not find any significant relation between BMI and antibody response. Our study is in line with recent publication describing that the BMI was not considered a factor associated with the antibody titre in the haemodialysis patients [21].

Despite the importance of vaccination, owing to the emergence of new variants and because of the vulnerability of this population, hygienic measures remain an indispensable cornerstone in the prevention of SARS-CoV-2 infections in dialysis units.

This study has several limitations. We acknowledge that our study only evaluated the immune response of the anti-Spike IgG antibodies, and it did not measure T cell response, which would help us to better understand the observed immune response. We did not identify patients with immunosuppressive therapy that could influence the immune response. Although the findings of previous studies regarding this factor were non-conclusive, it was observed that the lower respondents after the second dose had a history of immunosuppression [28,29], but, on the other hand, there are also studies that cannot find any suggestion that immunosuppression weakens this response [26]. In one study conducted by Bonelli, Michael et al. [30], they observed that B-cell-depleting therapy with rituximab affects the humoral immune response to SARS-CoV-2 vaccination and, furthermore, they also noticed a T-cell-mediated immune response even in B-cell-depleted patients. However, other studies suggested that immunosuppression therapy in kidney transplant patients is a strong inhibitor of humoral response [21,31].

## 5. Conclusions

Elderly patients on HD after the Comirnaty© vaccine developed a lower immune response than younger patients. Simultaneously, the levels of anti-Spike IgG antibodies declined faster between the third and sixth month. Moreover, heavier patients showed a greater immune response in both the vaccinated and control group, although without significance. Based on these findings, personalized immunization schedules of the elderly population may be needed to be considered for boosting to sustain immune protection.

## Figures and Tables

**Figure 1 ijerph-19-10091-f001:**
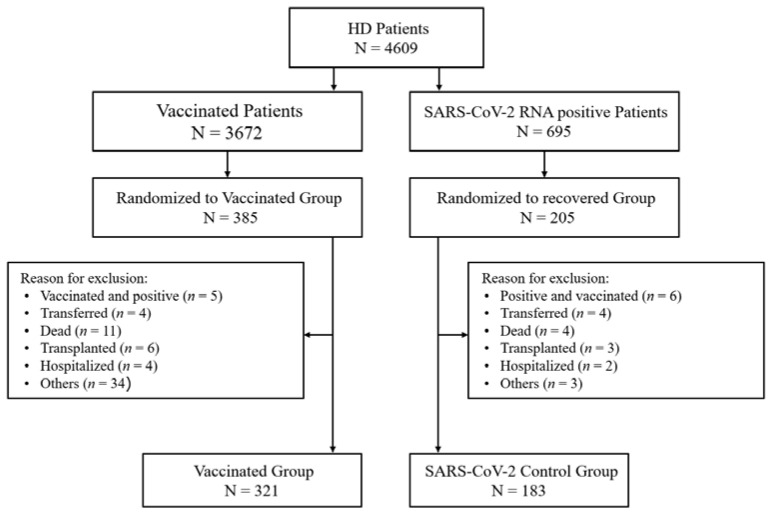
Patient flow diagram shows enrollment, randomization, and follow-up of study patients.

**Figure 2 ijerph-19-10091-f002:**
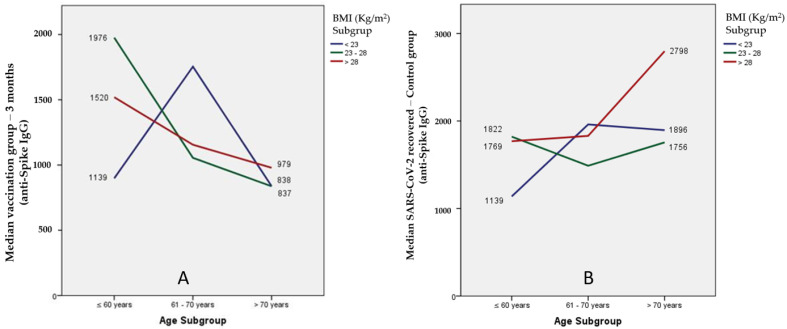
Graphic representation of anti-spike IgG antibodies level between weight and body mass index (BMI) variables in the vaccinated and control group. (**A**) In the vaccinated group, a greater variation of the humoral response was observed in the youngest and among the three sub-groups of the BMI variable. (**B**) There was a high humoral response in older (>70 years) and heavier (BMI > 28 kg/m^2^) patients in the control group.

**Table 1 ijerph-19-10091-t001:** Patient baseline characteristics of the vaccinated group and the COVID-19-recovered group.

	Vaccinated Group (*n* = 321)	COVID-19-Recovered Group (*n* = 183)	*p*-Value
Demographics			
Age (years), median (IQR)	67 [56–78]	70 [59–80]	*p* = 0.042
BMI (Kg/m^2^), median (IQR)	24.8 [22–28.20]	24.8 [21.50–29.30]	*p* = 0.646
Dialysis vintage (months), median (IQR)	62 [27.50–116]	57 [30–101]	*p* = 0.942
Male, *n* (%)	190 (59.20)	107 (58.50)	
Vascular access			
AVF *n* (%)	245 (66.0)	126 (33.9)	
AVG *n* (%)	30 (60)	20 (40)	
CVC *n* (%)	46 (55.42)	37 (44.58)	
Causes of ESRD (*n*, %)			
Diabetic nephropaty	49 (37.12)	83 (62.88)	
Hypertension/vascular disease	27 (41.54)	38 (58.46)	
Polycystic kidney disease	13 (37.14)	22 (62.86)	
Glomerulonephritis	22 (27.50)	58 (72.50)	
Chronic pyelonephritis	19 (44.19)	24 (55.81)	
Other causes	53 (36.31)	96 (64.43)	

Note: % presented are related to the frequencies evaluated within the respective class of the vaccinated and recovered group. For continuous variables, we present the median and the interquartile range (IQR). Mann–Whitney U test to compare the groups in which the variables were continuous. Abbreviations: AVF, arteriovenous fistula; AVG, arteriovenous graft; CVC, central venous catheter; BMI, body mass index.

**Table 2 ijerph-19-10091-t002:** Comparison of humoral immunity status between the two groups: vaccinated at 3(t0) and 6 (t1) months and COVID-19 recovered (control group).

Variables	Vaccinated Group t0	Vaccinated Group t1	COVID-19 Recovered Group	*p*-Value
Anti-Spike IgG (AU/mL) median (IQR)	1120 [493–2805]	455 [189–967]		<0.001 ^a^
Anti-Spike IgG (AU/mL) median (IQR)	1120 [493–2805]		1836 [749–5168]	<0.001 ^b^
Anti-Spike IgG (AU/mL) median (IQR)		455 [189–967]	1836 [749–5168]	<0.001 ^c^

Note: For continuous variables, we present the median and the interquartile range (IQR). Wilcoxon test to compare two paired groups was used for the vaccinated at the third and sixth months. Mann–Whitney U test to compare the groups in which the variables were continuous. ^a^ Wilcoxom = −15.597, *p* < 0.001; ^b^ Mann–Whitney = 22562, *p* < 0.001; ^c^ Mann–Whitney = 13391; *p* < 0.001. Abbreviations: AU/mL, arbitrary units per millilitre.

**Table 3 ijerph-19-10091-t003:** Comparison of humoral immunity status in the three age and body mass index subgroups.

Variables	Anti-Spike IgG (AU/mL) 3rd Month	Anti-Spike IgG (AU/mL) 6th Month	COVID-19 Recovered Anti-Spike IgG (Au/mL) 6th Month
Age Groups			
≤60 years, median (IQR)	1495 [686–3232]	554 [304–1224]	1769 [283–3394]
61–70 years, median (IQR)	1122 [482–2834]	416 [189–836]	1821 [990–4563]
>70 years, median (IQR)	874 [351–2053]	363 [128–852]	2096 [890–9849]
Body mass index			
<23 Kg/m^2^, median (IQR)	898 [342–2577]	363 [147–836]	1831 [794–6545]
23–28 Kg/m^2^, median (IQR)	1155 [544–2838]	493 [227–986]	1791 [515–3822]
>28 Kg/m^2^, median (IQR)	1229 [609–2853]	460 [196–1136]	2485 [818–5117]

Note: Three subgroups were created for age of ≤60, 61–70, and >70 years and for body mass index (BMI) of <23, 23–28, and >28 Kg/m^2^. We present the median and the interquartile range (IQR). Patients with BMI > 28 developed higher anti-Spike IgG levels in both vaccinated and control groups (median of 1229, IQR = 748–5168 AU/mL versus median of 2485, IQR = 818–5117 AU/mL). In the subgroup of vaccinated patients older than 70 years, fewer antibodies were observed when compared with the other subgroups. In contrast, in the control group, patients with more than 70 had better immune response (median of 2096, IQR = 890–9849 AU/mL). AU/mL, arbitrary units per millilitre.

**Table 4 ijerph-19-10091-t004:** Comparison of humoral immunity status in the age and body mass index subgroups.

Variables	Anti-Spike IgG 3rd Month (AU/mL)	*p*-Value	Anti-Spike IgG 6th Month (AU/mL)	*p*-Value	COVID-19 Recovered (AU/mL)	*p*-Value
Age Groups						
<70 years, (*n*) median (IQR)	(190) 1380 (641–3021)	0.002	712 (285–1774)	0.003	(95) 1807 (598–3945)	0.038
≥70 years, (*n*) median (IQR)	(131) 838 (345–2053)		684 (231–1949)		(88) 2096 (882–9872)	
Body mass index						
<30 Kg/m^2^, (*n*) median (IQR)	(268) 1083 (493–2813)		451 (196–981)		144–1817 (760–5330)	
≥30 Kg/m^2^, (*n*) median (IQR)	(53) 1210 (459–2607)	0.975	469 (166–835)	0.889	39–2621 (711–5166)	0.689

Note: Two subgroups were created for ages of <70 and ≥70 years and for body mass index (BMI) <30 and ≥30 Kg/m^2^. We present the median and the interquartile range (IQR). In the comparison between the subgroups, we used the Mann–Whitney U test. Abbreviations: AU/mL, arbitrary units per millilitre.

## Data Availability

The datasets used and/or analysed during the current study are personal health information obtained during provision of healthcare services and cannot be shared to protect their confidentiality in compliance with GDPR regulation.

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
