# Peer review of "Vaccination against SARS-CoV-2 in Haemodialysis Patients: Spike’s Ab Response and the Influence of BMI and Age"

_ijerph, 2022, doi:10.3390/ijerph191610091_

Round 1

Reviewer 1 Report

Review, Ponce et al., International Journal of Environmental Research and Public Health, 2022: Vaccination against COVID-19 in a haemodialysis clinic network. Spike’s Ab response at 3 and 6 months

Summary

In the study by Ponce et al, the group proposed to study in people with chronic kidney disease, the proportion of anti-Spike IgG after two doses of BNT162b2 mRNA vaccine. They showed that individuals, with a previous SARS-CoV-2 infection, developed a higher level of anti-Spike IgG compared to naïve individuals who had been vaccinated. In addition, they observed a decrease of these specific antibodies suggesting an attrition of the immune memory which can be influence by two factors: age and weight.

Overall, this manuscript focuses on a vulnerable population for which information on immune responses to vaccination is very important and this is also true in the context of COVID-19. However, the key messages are not well highlighted in the present manuscript, a lot of editing work is required before publication.

Major comments

Title

This title is not appropriate to your conclusions. Finally, we learn in the introduction that you talk about age and weight in addition of the Abs levels so you should be clearer in the title of your manuscript.

Abstract

1.     Your abstract is not well structured. You should remove the following terms “Background, Methods, Results and Conclusions”. In the Results part you never mentioned the weight.

Introduction

1.     You should introduce the role of the Ab in the immune response and modulate your introduction because you have focused only on the Ab production (only on the IgG, not IgA, IgM and IgD) not on all the humoral responses (ADCC, Neutralization, photype of B cells, …) nor on the cellular immune responses. What your observed is important to know, but this is a little part of the immune response to the SARS-CoV-2 vaccination.

Materials and methods

1.     This part is too dense, you should separate your different parts of the methods by paragraphs with title (as: Participants inclusion, Samples collection, Ig quantification, Measure of BMI, statistical analysis, etc …)

2.     You should generate a table with the clinical characteristics.

3.     It is extremely important and interesting to include a vaccinated cohort with a previous infection to SARS-CoV-2 to see how anti-Spike IgG evolved on these individuals. Why did you exclude them from your analysis? Nowadays, the vast majority of people have been infected with SARS-CoV-2 and are vaccinated even the people on hemodialysis, so how is their immune response to memory Abs memory?

Results

1.     This part lacks graphs to illustrate what you mention.

2.  Figure 2, authors should show individual values. Legend A and B are not legends.

3. Have you tried correlating the level of IgG with inflammatory markers in addition to age and weight? Inflammatory environment plays a role on the modulation of the immune response by the secretion of cytokines/chemokines, the impact on different T cell populations (such as the Tfh essential for B cell interaction and therefore the generation of Ab), the maturation of the B cells, …

Minor comments

1.     Throughout the manuscript you mixed the terms COVID-19 and SARS-CoV-2. COVID-19 is for the disease and SARS-CoV-2 is the virus responsible of the COVID-19 disease.

2.   Line 37-38 update your references (Goupil et al, 2021 10.1503/cmaj.210673). Four vaccines are allowed not three (J&J is missing)

3.     Line 50 add references on HBV and influenza vaccination.

4.     Correct throughout the manuscript “Spike” with an upper letter.

5.     Line 101, give the abbreviation of HD

6.  Throughout the manuscript keep the same vaccine name; Pfizer-BioNTech BNT162b2 mRNA, Pfizer BNT162b2, Comirnaty®, Pfizer vaccine or BNT162b2 mRNA vaccine but stay consistent

7.     Table 1 a lot of p-values are missing

Author Response

We attach a document with the answers

Thanks

Pont 1: Title

This title is not appropriate to your conclusions. Finally, we learn in the introduction that you talk about age and weight in addition of the Abs levels so you should be clearer in the title of your manuscript.

Response 1: We thank the reviewer for this suggestion. The Title was changed to:

“Vaccination against SARS-CoV-2 in haemodialysis patients: Spike’s Ab response and BMI and age influence”

Pont 2: Abstract

  1. Your abstract is not well structured. You should remove the following terms “Background, Methods, Results and Conclusions”. In the Results part you never mentioned the weight.

Response 2: Thank You for your observation. The headings “Background, Methods, Results and Conclusions” were removed from the abstract.

Regarding Body Mass Weight, the following sentence was added to the abstract: “Patients with higher Body Mass Index showed a greater immune response in both, the vaccinated and control group, however without significance”.

Pont 3: Introduction

  1. You should introduce the role of the Ab in the immune response and modulate your introduction because you have focused only on the Ab production (only on the IgG, not IgA, IgM and IgD) not on all the humoral responses (ADCC, Neutralization, photype of B cells, …) nor on the cellular immune responses. What your observed is important to know, but this is a little part of the immune response to the SARS-CoV-2 vaccination.

Response 3: We thank the reviewer for this suggestion. It was inserted: “Also, in dialysis patients, CD8+ T cells are diminished in both number and function. Since usually both arms of the immune system are impaired, could explain the increased susceptibility of this patients to viral and bacterial infections and their poor vaccination responses.”

Pont 4. Materials and methods

  1. This part is too dense, you should separate your different parts of the methods by paragraphs with title (as: Participants inclusion, Samples collection,Ig quantification, Measure of BMI, statistical analysis, etc …)

Response 4.1. Thank You for your observation. Several sub-sections were created as proposed

2.You should generate a table with the clinical characteristics

Response 4.2. We generate a table with the clinical characteristics (Table 1), unfortunately it is not possible to collect more data on these patients without the agreement of the ethics committee

  1. It is extremely important and interesting to include a vaccinated cohort with a previous infection to SARS-CoV-2 to see how anti-Spike IgG evolved on these individuals. Why did you exclude them from your analysis? Nowadays, the vast majority of people have been infected with SARS-CoV-2 and are vaccinated even the people on hemodialysis, so how is their immune response to memory Abs memory?

Response 4.3. We thank the reviewer for this commentary, but the protocol submitted to the ethics committee did not include the vaccinated cohort with a previous SARS-CoV-2 infection because at the time of vaccination, according to the scientific knowledge available, it was not expected that vaccinated patients would become infected after vaccination.

Furthermore, we also did not include it because in fact only a residual number of vaccinated patients had contracted the infection by the end of the follow-up period.

Pont 5. Results

  1. This part lacks graphs to illustrate what you mention.

Response 5.1. We appreciate the reviewer's suggestion, but we considered keeping the 4 tables and one figure, because we used SPSS and it is not the best program for graphing. Once again thank you for the suggestion

  1. Figure 2, authors should show individual values. Legend A and B are not legends.

Response 5.2. We appreciate the reviewer's suggestion and we considered keeping the information but as a note because it a useful summary information.

  1. Have you tried correlating the level of IgG with inflammatory markers in addition to age and weight? Inflammatory environment plays a role on the modulation of the immune response by the secretion of cytokines/chemokines, the impact on different T cell populations (such as the Tfh essential for B cell interaction and therefore the generation of Ab), the maturation of the B cells, …

Response 5.1. We thank the reviewer for this suggestion but we didn`t quantify inflammatory markers which prevented us from correlating them with IgG levels, and we consider this a limitation and we mentioned on the discussion that our study has several limitations

Pont 6. Minor comments

  1. Throughout the manuscript you mixed the terms COVID-19 and SARS-CoV-2. COVID-19 is for the disease and SARS-CoV-2 is the virus responsible of the COVID-19 disease.

Response 5.1. We thank the reviewer for this suggestion that improved the manuscript. We corrected the entire manuscript

  1. Line 37-38 update your references (Goupil et al, 2021 10.1503/cmaj.210673). Four vaccines are allowed not three (J&J is missing)

Response 5.2. The reference was updated

  1. Line 50 add references on HBV and influenza vaccination.

Response 5.3. Thank You for observation. In the maim body concerning this statement, references were introduced

  1. Correct throughout the manuscript “Spike” with an upper letter.

Response 5.4. Thanks, we corrected the entire manuscript

  1. Line 101, give the abbreviation of HD

Response 5.5. Thanks, it was given

  1. Throughout the manuscript keep the same vaccine name; Pfizer-BioNTech BNT162b2 mRNA, Pfizer BNT162b2, Comirnaty®, Pfizer vaccine or BNT162b2 mRNA vaccine but stay consistent

Response 5.6. We thank the reviewer for this suggestion that improved the manuscript. We corrected the entire manuscript

  1. Table 1 a lot of p-values are missing

Response 5.7. We thank the reviewer for this suggestion, but the remaining variables are nominal (yes or no) so we only present frequencies and percentages

Reviewer 2 Report

The manuscript was an interesting study on antibody levels in hemodialysis patients either with the SARS-CoV-2 vaccine or after recovering from infection. 

A few concerns that I have:

1) In section 3.3 and table 2, the SARS-CoV-2 recovered group's antibody level is compared to the vaccinated group at t0; however, the recovered group levels were measured six months after recovery and should be compared to the vaccinated group at t1. The authors show that antibody levels decrease over time, and the comparison between the two time points might be problematic since it can be assumed that closer to the infected time, the antibody level would be higher. 

2) There were a lot of grammatical issues in the manuscript that made reading and understanding difficult and should be addressed before publication.

Overall the manuscript is of interest and would add to the overall scientific knowledge on SARS-CoV-2 immune response. 

Author Response

We attach a document with the answers

Thanks

Response to Reviewer 2 Comments:

The manuscript was an interesting study on antibody levels in hemodialysis patients either with the SARS-CoV-2 vaccine or after recovering from infection.

Pont 1: 1) In section 3.3 and table 2, the SARS-CoV-2 recovered group's antibody level is compared to the vaccinated group at t0; however, the recovered group levels were measured six months after recovery and should be compared to the vaccinated group at t1. The authors show that antibody levels decrease over time, and the comparison between the two time points might be problematic since it can be assumed that closer to the infected time, the antibody level would be higher. 

Response 1. We thank the reviewer for this suggestion, but the Table 2 shows all possible comparisons: T0 vs T1; T0 vs Recovered group and T1 vs Recovered group. Thus, we compared the recovered group levels (six months) with the vaccinated group at t1 (six months). However, we were not able to assess antibody levels at 3 months after SARS-CoV-2 infection.

Pont 2: 2) There were a lot of grammatical issues in the manuscript that made reading and understanding difficult and should be addressed before publication.

Response 2. We thank the reviewer for this suggestion that improved the manuscript. We corrected the entire manuscript
